# Planning and Implementation of COVID-19 Isolation and Quarantine Facilities in Hawaii: A Public Health Case Report

**DOI:** 10.3390/ijerph19159368

**Published:** 2022-07-30

**Authors:** Victoria Y. Fan, Craig T. Yamaguchi, Ketan Pal, Stephen M. Geib, Leocadia Conlon, Joshua R. Holmes, Yara Sutton, Amihan Aiona, Amy B. Curtis, Edward Mersereau

**Affiliations:** 1Social Science Research Institute, College of Social Sciences, University of Hawaii at Mānoa, Honolulu, HI 96822, USA; ctly21@hawaii.edu (C.T.Y.); ketanpal@hawaii.edu (K.P.); smgeib@hawaii.edu (S.M.G.); leocadia.conlon@hawaii.edu (L.C.); 2Hawaii State Department of Health, Behavioral Health Administration, Honolulu, HI 96819, USA; jholmes@dhs.hawaii.gov (J.R.H.); yara.sutton@doh.hawaii.gov (Y.S.); amihan.aiona@doh.hawaii.gov (A.A.); amy.curtis@doh.hawaii.gov (A.B.C.); edward.mersereau@doh.hawaii.gov (E.M.)

**Keywords:** COVID-19, isolation and quarantine, containment and mitigation strategies, homelessness

## Abstract

In response to the second surge of COVID-19 cases in Hawaii in the fall of 2020, the Hawaii State Department of Health Behavioral Health Administration led and contracted a coalition of agencies to plan and implement an isolation and quarantine facility placement service that included food, testing, and transportation assistance for a state capitol and major urban center. The goal of the program was to provide safe isolation and quarantine options for individual residents at risk of not being able to comply with isolation and quarantine mandates. Drawing upon historical lived experiences in planning and implementing the system for isolation and quarantine facilities, this qualitative public health case study report applies the plan-do-study-act (PDSA) improvement model and framework to review and summarize the implementation of this system. This case study also offers lessons for a unique opportunity for collaboration led by a public behavioral health leadership that expands upon traditionally narrow infectious disease control, by developing a continuum of care that not only addresses immediate COVID-19 concerns but also longer-term supports and services including housing, access to mental health services, and other social services. This case study highlights the role of a state agency in building a coalition of agencies, including a public university, to respond to the pandemic. The case study also discusses how continuous learning was executed to improve delivery of care.

## 1. Introduction

The coronavirus disease 2019 (COVID-19) pandemic, with much uncertainty of the trajectory at the start, created many challenges to effectively mitigate the spread of COVID-19 [1]. The surging nature of COVID-19 cases challenged the capacity to respond to and control the spread across the three epidemiologic pillars of testing, tracing, and isolation [2,3]. Testing individuals is vital to detect the presence of COVID-19. Tracing identifies those who have been exposed. Isolation and quarantine ensure that all parties prevent the further spread of the disease.

Starting 10 August 2020, the Hawaii State Department of Health Behavioral Health Administration (DOH BHA) was assigned the responsibility to lead the isolation and quarantine function to further contain and mitigate the spread of COVID-19 in the major urban center of Honolulu, Hawaii. Prior to the involvement of the DOH BHA in isolation and quarantine services, these services had been the domain and responsibility of the Disease Outbreak and Control Division, as well as of the Office of Public Health Preparedness in the DOH, units that are in separate administrations from the BHA. Thus, DOH BHA’s role in providing isolation and quarantine services was arguably novel in a state organization chart that has been long fragmented by biomedical categories [4,5]. Recognizing the challenges of surge capacity, the DOH BHA led, coordinated, and contracted a coalition of public and private partnering agencies including the City and County of Honolulu, the University of Hawaii, and several other agencies.

Other countries have utilized facilities such as hotels as an avenue toward rapid mitigation for the spread of COVID-19 [6,7,8] along with alternative care sites away from usage of hospital beds capable of medical-related services [9,10]. The COVID-19 pandemic also revealed the disparities of vulnerable populations, such as individuals experiencing homelessness, in the process of implementing isolation and quarantine facilities [11]. In one study in San Francisco that analyzed a hotel used for isolation and quarantine from 19 March to 31 May 2020, 50% of 1009 hotel participants experienced homelessness [12]. People with COVID-19 may experience behavioral health challenges, as well as other social problems. In the response to COVID-19, historical organizational fragmentation separated COVID-19 and infectious disease response from the behavioral health response, even though patients and the public, obviously, can experience both simultaneously. Thus, in this narrative case study, we also focus on the ways in which behavioral health approaches are integrated with COVID-19 responses. With a focus on homelessness, behavioral health, and the vulnerable populations in the communities, DOH BHA incorporated connections within the isolation and quarantine infrastructure to connect individuals to needed services for mental health, substance use, and housing.

Different strategies to use hotels as facility placement locations were adopted in other areas in the United States, e.g., in New York [13], as well as in other countries including Australia, Italy, and Spain [8,14,15]. Quarantine hotels served as a temporary premise toward quickly setting up an infrastructure that provided a safe and private space to self-isolate. In some countries, e.g., Spain, the majority of those isolated and quarantined in hotel facilities were travelers from abroad [8]. Local, state, and national guidelines for COVID-19 isolation and quarantine were constantly evolving and changing. In this narrative and historical case study, we focus on the isolation and quarantine facility system in one state jurisdiction (Hawaii) that is primarily for residents rather than travelers.

Proper coordination and communication are vital toward effectiveness and efficiency of newly built systems for containment of COVID-19, as shown in Singapore [7]. Communication and coordination, too, were crucial components of focus for the DOH BHA when setting up system infrastructure. There were several functional teams that needed to communicate effectively, both communication internally (across teams and within teams) and externally to other outside stakeholders and partners. It is common for complications and complexities to emerge when addressing COVID-19, making communication among several parties, or cross-functional communication and coordination, very important [16]. This case study reflects on the implementation experiences in one jurisdiction by applying the plan-do-study-act (PDSA) improvement model and framework to review and summarize the implementation of this system.

## 2. The Plan-Do-Study-Act Improvement Model and Framework

The PDSA model is a standard reference model for quality improvement [17,18]. The “plan” stage represents making predictions, goals, and tasks to be assigned (Figure 1). The “do” stage is implementing this plan. The “study” stage is analyzing the workflow or data to understand if changes need to be made. Last, the “act” stage is to make and implement changes based on what was found in the ”study” stage. The PDSA model provided DOH BHA a framework that allowed for flexibility throughout the process of designing and implementing various activities of this system.

### 2.1. Plan

#### 2.1.1. Vision and Philosophy

Early on in August 2020, DOH BHA leadership recognized that no single agency had the capacity to provide all services needed for a comprehensive system of isolation and quarantine services encompassing transportation, food services, hotel, call center, case management, and so on. DOH BHA further recognized that the capacities developed during the pandemic could create “durable assets” that could benefit the public behavioral health system even after the pandemic’s resolution. DOH BHA established the priority and vision to address not only immediate COVID-19 needs but also ensure linkage to the continuum of care, including access to housing, behavioral health, mental health and substance use services, and other social services. Thus, DOH BHA recognized coordination and partnership of private and public agencies as essential. As DOH BHA had a structure already in place to address behavioral health, mental health, and substance use disorders, the vision was to incorporate and expand upon the existing DOH BHA infrastructure with the capabilities of handling isolation and quarantine for COVID-19 cases. DOH BHA established standards of what the system would provide including: (1) use of a call center to accept requests (Hawaii CARES), (2) 24/7 operation, (3) client placement in a hotel facility within four hours of client’s initial request, and (4) providing isolation and quarantine services to a benchmark of 9% of all COVID-19 cases, as 9% of Hawaii’s population lived in shared bedrooms and could not safely isolate [19].

#### 2.1.2. Key Planning Tasks

The DOH BHA was given less than one-week’s notification to plan the system. Despite the short time frame, the DOH BHA gathered a team of existing partner agencies through which it already had existing contracts that could be modified and adapted for expanded scope of services for COVID response for isolation and quarantine. DOH BHA was able to contract with private agencies under the emergency orders authorized by the governor, as well as the emergency authorizations under public procurement law. DOH BHA’s plan was then to create an online intake form and central call line that allowed for a triage team to appropriately connect clients to needed services, as well as to connect individuals to case managers onsite if a facility placement is needed that would then handle check-ins, daily wellness checks, and discharge (Figure 2). DOH BHA also drew on past experiences including those from March 2020 when DOH BHA established the state’s first stand-alone temporary quarantine facility for homeless individuals [20].

#### 2.1.3. Surge Capacity in Administration, Staffing, and Contracting

DOH BHA relied on existing infrastructure to meet surge capacity. Hawaii CARES, an existing 24/7 call center for mental health and substance use disorders, was designated as the means to accept requests for isolation and quarantine. DOH BHA also relied on extending and modifying existing contracts with partnering agencies, rather than create new contracts. Finally, DOH BHA nimbly increased staff capacity through Hawaii CARES by the hiring of team members, including student employees, using the infrastructure of a public corporation (Research Corporation of the University of Hawaii) as part of the flagship public university system, the University of Hawaii, which was also skilled in the information technology, evaluation, and training needed to operate a call center.

#### 2.1.4. Roles and Responsibilities

With the structure of the workflow set in place, various functional teams were made to serve specific functions and hold different responsibilities among each other. Table 1 presents the functional teams as follows: the triage team, care coordination team, hotel team, transportation team, food and supply delivery team, and the testing team. Each functional team had unique functional roles and responsibilities (Table 1).

### 2.2. Do

#### 2.2.1. Request and Intake Form

The implementation of the isolation and quarantine placement system depended on a unified entry gate for isolation and quarantine placement through a 24/7 call center. Callers seeking isolation and quarantine were triaged to an operator who could assist the caller to complete an isolation and quarantine request Qualtrics mobile-friendly web form. Qualtrics mobile-friendly web form, an online survey, was the available survey tool to the operations of the isolation and quarantine services during the emergency period for which existing security protocols had previously been signed. The form was revised multiple times throughout the emergency period to best suit the needs of the operations for delivering isolation and quarantine services. The exact number of edits to individual questions, question order, adding new questions, and deleting old questions was numerous and thus not counted due to the continuous nature of the operations requiring real-time feedback.

Due to the need for timely response, the coordination team had to rely on whatever existing information technology infrastructure was available to them. This web form was then filled by healthcare and social service providers, the call center operators, and community members who guided them to specific forms depending on whichever role they had indicated at the beginning of the survey. Given the option at the end of the survey, anyone could provide feedback in an open-ended field about the form and workflow, which was then discussed through daily and weekly meetings with the leadership team that oversees the workflow consisting of DOH BHA administrators, clinic manager of the call center, leads for each team of the workflow, and stakeholders. The form was revised to be made as concise as possible.

#### 2.2.2. Client Tracking System

The data system used for tracking clients throughout the whole workflow process—request, triage, transportation, hotel, food, and discharge—evolved, initially starting with a Microsoft Excel sheet in a Microsoft Teams and later in December 2020 a Microsoft Dynamics 365 database.

#### 2.2.3. Triage

Initially, the Hawaii CARES isolation and quarantine care coordination team pulled data from the webform into the Excel tracking system. Triage team members then looked at newly added cases and determined epidemiologic eligibility for services, including whether the individual could not safely isolate at home, whether the individual was positive or awaiting test results, and so on, based on CDC guidance. The triage team then followed-up with the client and their entire profile to ensure proper placement. If the triage team deemed the client to need isolation and quarantine placement at an external facility, then the client case would be assigned to the hotel team, which viewed their profile to determine which facility best suited the individual profile and level of care needed (e.g., based on activities of daily living).

#### 2.2.4. Client Connection

Once a facility was assigned to the client, the care coordination team informed the client of their assignment and how to check-in to the facility. If needed, the care coordination team also arranged transportation for the client, by communicating with the DOH BHA-contracted transportation team using encrypted communication modality. The driver then securely transported the clients from the pickup location to the assigned facility, where the client was connected to an assigned case manager and entered isolation and quarantine.

#### 2.2.5. Screening, Support, and Discharge

The case manager would then screen to see if further services for medical or social problems were needed and would perform daily wellness checks and deliver meals three times a day. Upon completion of isolation or quarantine, case managers would then arrange transportation if needed and handle discharge. The triage team also facilitated testing if the client requested COVID-19 testing and were determined to be eligible. Eligible clients isolating or quarantining at home could request the triage team for food and supply delivery services to the client’s home, delivered by multiple partnering agencies.

### 2.3. Study

Throughout the implementation period, there was continuous learning of how to develop, change, and revise the workflow for improvement. Daily meetings were held to review the adequacy of the form and issues that may have occurred while placing individuals into isolation and quarantine. After consensus and if needed, the form would be revised and edited immediately after the daily meeting. Some of the learning that took place was to reduce confusion in patient placement, ensure timeliness in placement, and address specific case-by-case issues for patient placement that arose upon specific patient requests.

Addressing the needs of vulnerable populations was a priority. Linguistic and cultural barriers challenged the ability of the system. To better support the linguistic needs of clients, the workflow was adapted through a partnership with a local community-based organization, We Are Oceania, as well as a national line to support translation. Linguistic and cultural competency continue to be an important area for capacity development for improved preparedness.

Many individuals in need of isolation and quarantine experienced homelessness and behavioral health complications. Qualified case managers were stationed at each facility with each client screened to see if additional services were needed. Ongoing feedback from exit interviews was also used to improve the quality of services provided.

Meetings with the functional teams, as well as meetings with external stakeholders, provided feedback for improvement throughout the implementation of the system. More than 3248 individuals between 12 August 2020 and 10 December 2020 were connected to isolation and quarantine, food, and/or testing services, and the target of providing services to individuals that exceeded more than 9% of all COVID-19 cases in the state was reached [21]. Once gaps in client placement or workflow had been identified through feedback from the intake form or staff, changes were then implemented within the system and quickly adjusted through daily meetings with those involved with client placement and the leadership team to ensure open communication, feedback, and improvement. Daily meetings ensured open communication, feedback, and improvement.

### 2.4. Act

The act stage consisted of implementing new changes based on what was found throughout the study stage (Figure 1). Staffing and technology were persistent enablers of, as well as challenges to, implementation. Staffing was initially limited and relied on staff overtime, which resulted in staff fatigue and burnout. The capacity to rapidly recruit new staff was severely stressed. Meeting surge capacity could be more easily tackled with the creation of a reserve roster of individuals who could be easily tapped for deployment. The reliance on student employment and other forms of temporary and casual hires were instrumental to ensuring rapid stand-up of the care coordination team to operate on a 24/7 basis.

The use of the Excel spreadsheet became challenging as software was unstable with multiple concurrent users working on it at one time, potentially deleting another individual’s work if multiple people are working on the same cell. The transition to the Microsoft D365 system as a formal database helped ensure data stability through preventing these unexpected deletions from occurring and later reduced the care coordination team role. The changes were not implemented immediately, but progressively as communication with many teams took place.

## 3. Discussion

This article is a narrative case study that, like many qualitative case studies, lacks a comparison or counterfactual to help shed light on what might have happened in the absence of this program. Nevertheless, we note that prior to the expansion of isolation and quarantine hotels, the jurisdiction had only 60 beds procured for the city prior to DOH BHA implementing this system. Under DOH BHA leadership, up to 400 beds were procured and a system of coordination was established to operate the isolation and quarantine for smooth placement of clients.

Future preparedness for rapid implementation of a system can be improved through staffing reserve capacity, technological capacity, and strategic collaboration with community-based organizations with linguistic and cultural competencies that address the needs of vulnerable populations.

Hawaiʻi is a multicultural place where many different languages are spoken. Linguistic barriers have long challenged healthcare access, compounded by a lack of cultural competency of providers [22,23]. Enhancing language access can be done by building upon existing communication infrastructure between provider organizations, such as those agreements that had already existed prior to the pandemic, or the existing call center communication infrastructure that had also been in place since 2015. Through partnerships with those who are capable of addressing the linguistic barriers associated with communication problems, this promoted an increase in appropriate coordination for culturally diverse individuals and families.

The system benefited from using and building upon existing communication infrastructure, and a 24/7 call center specifically, for greater coordination during a pandemic [24]. While creating new communication infrastructure was an option, it became apparent that using an existing call center was more efficient and cost-effective. As cases rose, the capacity to withstand the excess call volume became difficult to maintain at the call center until more call operators and other team members were recruited and hired into their respective roles. Communication among different teams became vital to ensure stability of infrastructure, allowing for timely and prompt connections to be made utilizing a key element of feedback loops [25].

This case study also emphasized the role of a public university in managing and staffing a 24/7 call center that served as the care coordination team and central coordination hub for operations, working in close partnership with the health authority. When state government hiring infrastructure is limited in its ability to hire nimbly, the use of contractors including public universities can enable and ensure adequate staffing for a project that requires surge capacity [26].

Use of feedback loops helped to improve the system through regular meetings internally and externally, as well as with quality assessments and exit surveys of clients who were discharged. Meetings were especially important when new problems emerged, or if there was a novel incident that needed to be addressed for which existing protocols were not sufficient. Constant review of the system with those coordinating the day-to-day operations and administration allowed various events and complicated obstacles to be quickly addressed and solved.

With the intricacies of building a system infrastructure, simple communication with policymakers and community members was crucial to implement a new intake system [19]. Moreover, the pandemic has exacerbated needs for mental health [27,28]. As the system described herein relied on behavioral health infrastructure, it was thus possible to address mental health disorders in a timely manner to connect individuals and families to further needed services, leading to easy access of behavioral health after resolution of their COVID-19 concerns.

This implemented system described herein is notable for its role as a state behavioral health authority playing a role in COVID-19 activities, reflecting the potential to rely on partners outside of a traditional infectious disease or emergency medicine framework. By leveraging the public behavioral health system for a health response, this case study demonstrated the ability to mitigate the COVID-19 pandemic, as well as concurrent mental health issues [21]. As such, surge capacity can be met through a broad coalition of willing partner agencies led by a state agency with existing capacity to quickly respond and provide a comprehensive system of care encompassing a variety of functions including request and intake, triage, transportation, facility placement, case management, and discharge planning. This case study emphasizes the role of a public university to provide critical staffing and enhance surge capacity during a pandemic. Future research is needed to understand best practices and lessons across countries for developing and preparing surge capacity before a pandemic hits. 

Future research would benefit from a comprehensive review of studies and experiences about isolation and quarantine hotels around the world. In particular, future research is needed to understand the ways in which behavioral health was integrated and coordinated as part of the COVID-19 response and the ways in which organizational structures and siloes hindered or helped to integrate these two challenges. This narrative historical case study is one innovative though small contribution. 

## 4. Conclusions

This case study reviews the PDSA model used to implement a COVID-19 isolation and quarantine facility system with the public behavioral health system as its foundation, as well as lessons learned and the implementation plan of one city in the United States that were not exclusively clinical or laboratory focused.

Like many cities across the United States, Honolulu has been facing serious social challenges of homelessness and housing, mental health, and substance use, further exacerbated by the challenges of a COVID-19 pandemic and other infectious diseases, which have been a long-standing concern of Hawaii as a tropical location. There is a need for timely and accurate communication and coordination to address a variety of problems, not only COVID-19. This paper further contributes to the literature about the role of programs that address the social, economic, and psychological underpinnings that contribute to the risk of infectious disease spread.

## Figures and Tables

**Figure 1 ijerph-19-09368-f001:**
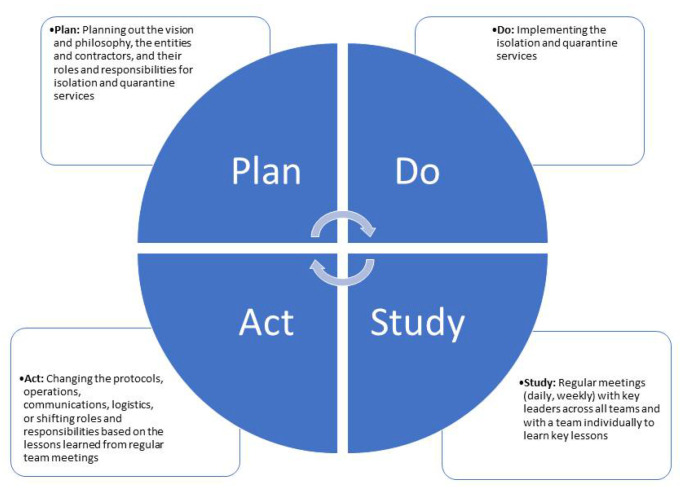
PDSA Cycle. Notes: the PDSA cycle was adapted and utilized for the purpose of creating and implementing a system that allowed for a structured quality improvement regime and allowed for constant analysis and changes.

**Figure 2 ijerph-19-09368-f002:**
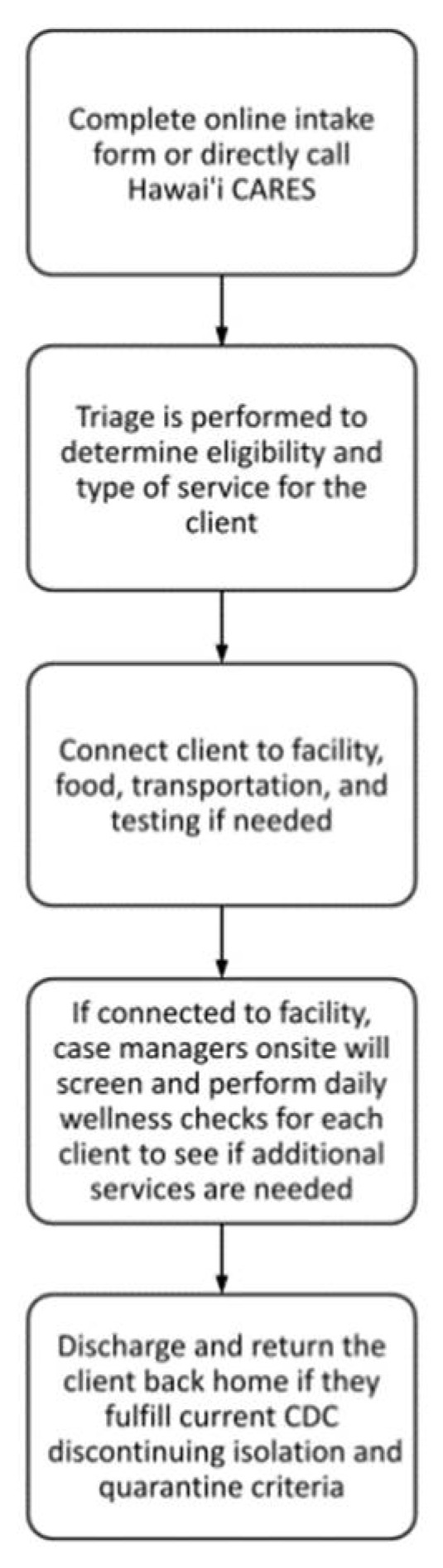
Workflow diagram summary of the key planning tasks of the Hawaii isolation and quarantine system, 10 August 2020 to 30 September 2021. Notes: The isolation and quarantine system starts with the intake form. The triage team then connects them to the needed services accordingly. Upon arrival at the facility, case managers screen each client and throughout their stay perform daily wellness checks to determine if additional services are needed. Discharge is performed if the client fulfills the current CDC guidelines for discontinuing isolation and quarantine criteria.

**Table 1 ijerph-19-09368-t001:** Roles and Responsibilities of Key Functional Teams in the Isolation and Quarantine System, 10 August 2020 to 30 September 2021.

Functional Team	Roles and Responsibilities
Triage Team	Triage cases that have been inserted into the tracking system by the care coordinators (determine what type of service client is requesting, determine if client is appropriate for services, etc.) Verify guest checkouts Follow-up with clients if needed
Care Coordination Team	Input cases into tracking system submitted from the Qualtrics form Communicate with the call center if further questions about a case Monitor and respond to publicly available email Assist triage team when needed Schedule transportation for applicable clients that request it Follow-up with clients if needed
Hotel Team	Place applicable clients at appropriate hotel Conduct behavioral health consults to see if appropriate for hotel
Transportation Team	Pick-up clients based on order of request Drop-off clients at designated facilities Communicate with care coordination team to determine client information, requested pick-up time and address Communicate with clients to ensure a smooth transition into the vehicle Communicate with case management at destined location to be ready for a drop-off or pick-up
Food and Supply Delivery Team	Work on cases in the order they are imputed into the tracking system Communicate with clients to confirm personal information, address, number of individuals needing food and supply, allergies, and types of supplies needed Deliver food and supply for clients to their designated address
Testing Team	Work on cases in the order they are imputed into the tracking system Communicate with clients to confirm personal information, address, number of individuals needing a test, and when to expect the test to arrive

Notes: The isolation and quarantine system was initially started with six teams: triage, care coordinators, hotel, transportation, food and supply delivery, and testing. Each team had their respective responsibilities, including the list herein.

## Data Availability

Not applicable.

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
