# Peer review of "Planning and Implementation of COVID-19 Isolation and Quarantine Facilities in Hawaii: A Public Health Case Report"

_ijerph, 2022, doi:10.3390/ijerph19159368_

Round 1
Reviewer 1 Report
This is a case study of planning and implementation of COVID-19 isolation and quarantine facilities in Hawaii from March 2020 to December 2020. The case it reports on is unique in coordinating resources in both the public and private sector. The conceptualization provided in the case study regarding both the items involved in the workflow and the role and responsibilities of the key teams is helpful to policy makers. However, the way in which the case study is presented is vague, leaving many important questions unanswered. As well, the writing style is often colloquial, using words in a way that may confuse readers. Sometimes, grammar is sometimes a problem.
To be publishable, this case study needs to answer these questions and be reorganized to include the roles and responsibility as part of the section on planning. Furthermore, if behavioral health is a key component of the intervention—as is stated in the paper—then this needs to be mentioned in both the introduction and the conclusion and more information provided on this focus in the remainder of the case study.
Line by line suggested edits.
1 Change “Article” to “Case Study”.
16 Change “article provides” to “is”.
18 Change “article” to “case study”.
27 Please introduce COVID-19 and explain why it is a public threat requiring these measures for isolation. As well, always refer to COVID-19, not just COVID.
28 There needs to be information regarding for what testing, tracing and isolation relate to regarding the three pillars.
33 How was the BHA, as a public institution, able to coordinate a coalition that contained private agencies?
39 Change “was” to “were”
40 “This study was done”, which study do you mean? Reference 8 does not refer to a study done in San Francisco.
47 Change “Australia, Italy and Spain [5, 10, 11]” to “Australia, Italy and Spain [10, 11, 5]”.
52 Change “while in” to “while those in”.
59 Change “typical of” to “typical for”.
70 Change “COVID” to “COVID-19”.
77 Change “COVID” to “COVID-19”.
80 Change “a one-week” to “one-week’s”.
81 How was the BHA able to gather the team in in short time period?
84 Change “including from” to “including those from”.
89 What is “plussing up”? Do you mean “extending”, “increasing” or something else?
91 Change “members which included” to “members, including”.
92 Move Figure 1 to follow section 2 rather than after section 3 so that it is closer to where it is mentioned.
93-95 This is part of the instructions to authors from the MDPI template that the author has forgotten to remove. Please remove it.
100 “Qualtrics mobile-friendly web form” please provide a description of this form and the reason this form was chosen.
101 “This request could also be completed directly by health care and social services providers.” If the health care or social services providers completed the request did they use the Qualtrics mobile friendly web form? Was there any difference between how the caller completed the form and how the providers did?
102 “received continuously” How was this information received continuously?
104 How many times was the form revised?
Figure 1 There are five areas of information as part of Figure 1. Each one of the subsections of section 3 should related to one of these five areas. As the subsections currently are focused, there is no information on either the daily wellness checks or the discharge and return—the second last and last areas on the workflow diagram.
139 What do you mean exactly by “continuous learning”? Describe what went on specific to this learning.
148 Change “were” to “was”.
150 How often did these meetings take place and who all was in attendance?
152 Change “3 248” to “3,248”.
154 Change “COVID” to “COVID-19”.
155 How was the workflow continually adapted and by whom?
156 Who was involved in the daily meetings?
163 In what way was Excel “unstable”? Are you sure it was the software that was unstable and not the way it was utilized by the users?
164 In what was the Microsoft D365 system more stable than Excel? Why was the Microsoft D365 system chosen if information could not be shared?
168 This is the first time it is stated that there was an emphasis on behavioral health of this system. This information should be mentioned under 2.1 Vision and Philosophy.
172 Change “mental health and the COVID-19 pandemic” to “the COVID-19 pandemic as well as concurrent mental health issues”.
Table 1 Place the dots closer in front of each of the points under Responsibilities and reduce their size. Table 1 is not referenced anywhere in the text. These roles and responsibilities should be discussed under a subsection called “2.4 Roles and Responsibilities” and the table referenced then.
182 What exactly is meant by “stronger linkages”?
196 Change “amounts of phone calls” to “amount of phone calls”.
197 Change “onboarded into” to “assumed”.
202 Change “were either recently onboarded” to “joined later”.
203 Change “or was” to “or there was”.
204 Change “day to day and” to “day-to-day operations and”.
207 Change “understand in simplistic terms” to “understand what is involved in simply terms”.
214 Change “COVID” to “COVID-19”.
219 Change “quarantine system” to “quarantine system with a focus on behavioral health”.
Author Contributions: How is the conceptualization provided from V.Y.F. (line 220) different from the visualization provided by V.Y.F. (line 223)?
References: All references need to be redone in the MDPI style as provided on the IJERPH template.
Reviewer 2 Report
Comments
This text looks more like a case study than an article, which is scientifically important so that the academic community can get to know the reality of all countries in the world. The reason is simple: the regions are inhabited by humans. What was done in Hawaii was not very different from the reaction in other countries, although with different logistics, excluding the effect of scale and population density. Regardless of the geographic area, in any territory something had to be done. However, the means and leadership of the process were unequal to mitigate the pandemic, either for organizational reasons or for cultural and even political reasons.
Language barriers already existed before the pandemic. Most countries were caught off guard with the pandemic caused by the SARS-CoV-2 virus, responsible for the Covid-19 disease, which brought to the earth's surface the inability to respond and prepare for a global pandemic that, not being historically unique, it could happen. There were warnings, but preparation for such an eventuality was nil. In fact, there had already been epidemics this century, e.g., in 2003 (outbreak of coronavirus in about 12 countries and which took about nine months to be declared extinct) and in 2009 (influenza A).
NOTE: The sentence on page 120 is missing a full stop.
Suggestions
Language barriers already existed before the pandemic. And what was it like for populations to live with this before this “war without sound”? How did communications infrastructure work before the pandemic? Why haven't they evolved enough? These are two suggestions that I leave to the authors to include a short historical framework in the article, which will not be difficult to conceive.
Periodically, pandemics occur all over the world and, therefore, the need for responses to face them. Prevention is the best strategy. Probably this will not be the last pandemic to face. History teaches that this type of outbreak is a reality we have to live with. Even the interval between epidemics with potential pandemic effects may become shorter and shorter. In this sense, the Director-General of the World Health Organization (WHO) warned that all countries should invest in public health. This is a global security issue that many countries do not understand or underestimate, and which must become an urgent and ongoing priority. Here's another suggestion: what Hawaii is doing to mitigate an almost guessed pandemic eventuality. What challenges does Hawaii face in terms of housing, communication infrastructure, mental health and homelessness in the event of a pandemic situation? Now, this is another suggestion I make to add to the article.
I have another suggestion. It is true that viruses are part of the ecosystem in which we live, and it is impossible to eradicate them from the planet. However, a new, more sustainable social and economic policy are key parts of the world economic-industrial architecture that, if suppressed with clean technologies, reduce the statistical risk of imploding new infectious foci. How are public authorities in Hawaii concerned about this predictable global phenomenon? Have you analyzed them?
To be well prepared, the first step is to assess what has been done and how it has been done, so that a script can be established for dealing with an epidemic or pandemic. Only in this way is it possible to envisage that there will be established health structures, equipped and prepared with sufficient, properly trained personnel, with a population vision and not exclusively clinical or laboratory. This is a correction that I leave to the authors as a complement to the previous suggestion.
Reviewer 3 Report
Review of Planning and Implementation of COVID-19 Isolation and Quarantine Facilities in Hawaii: A Public Health Case Report
1. Please mention figure 1 and table 1 in the text before they appear.
2. Particularly this article presents a descriptive observation of how Hawaii have implemented its COVID-19 countermeasures. However, it is just a description of it. This reviewer lacks more methodology to be used with the data already existent.
3. For example, no results are indicated or can be proved if these countermeasures were enough or not
4. No results about how these individuals were threatened during the quarantine period and its outcomes;
5. Another point refers to the period this research is being presented, because as of early 2021, hotels were being used around the world seeking to isolate individuals and many data were generated with numerical results about it. Up to now, at least, researches about it should compile numerical results as reviews, or at least as meta-analysis. This research in particular presents no comparisons, methodically, speaking.
6. Rather than demonstrating “lessons”, I believe the research is more presenting their method “Plan-Do-Study-Act (PDSA)”
7. Here in line 217, “This experience was unique for its leadership of a public behavioral health system to organize a partnership of agencies to rapidly implement the isolation and quarantine system”…, was is good or bad? And how do you prove it?
8. There is no limitation of the study. How authors compared PDSA with other policies in the same country or region, or if all the same, other countries? This question leads again to other question: Why PDSA proved to be right in terms of efficacy, efficiency and effectiveness?
Round 2
Reviewer 1 Report
Thank you to the authors for their revisions. They have greatly improved the paper. However, there is still much to do make this study publishable. If the change provided by the authors is not mentioned in what follows, it means that the revision provided by the authors is accepted. Thank you for making these changes.
In the authors’ cover letter response, it would have been helpful for the authors to indicate each change that was made in response to points that had been made by the reviewer, rather than only commenting on the larger suggestions that were made. That way, the reviewer would have known upon reading the cover letter whether the suggested changes were made without having to search for them.
One of the problems that remains with this submission is the understated importance of “the Plan-Do-Study-Act (PDSA) improvement model and framework to review and summarize the implementation of this system”. Although the PDSA is the model and framework for this work, the paper is not structured to recognize the significance of the PDSA. Instead, there is no information on the PDSA in itself and “Plan” has one subsection, “Do” has another and “Study and Act” share a separate section. In place of this arrangement. “The Plan-Do-Study-Act Improvement Model and Framework” should be the section 2 title. The first paragraph of this section should explain the model, citing references, and why the PDSA is both a model and a framework (citing references) for the work undertaken in Hawaii.
At this point, a new Figure 1 should be created, then cited and displayed, indicating the Hawaii PDSA. The new Figure 1 should have four elements—“Plan”, “Do”, “Study”, and “Act”. Within the elements (which can be displayed in a manner similar to the elements in Figure 2) would be the current subsections listed in the paper. For example, the “Do” element of the new Figure 1 should include “Vision and Philosophy”, “Planning Tempo (please see information below about changing this subsubheading)”, “Surge Capacity in Administration, Staffing, and Contracting”, and “Roles and Responsibilities”. Then, in the main body of the paper, there should be separate subsections for each of “Plan”, “Do”, “Study”, and “Act” to relate the information in Figure 1. Each subsection should begin with a definition of the terms “Plan”, “Do”, “Study”, and “Act”, citing examples from references. With this background, the current subsections in the paper would become subsubsections under each of these headings of section 2—“Plan”, “Do”, “Study”, and “Act”. What this will mean is that “Study” and “Act” will be given equal representation to “Plan” and “Do”, unlike how the paper is structured now.
Beyond these general comments, the following represent the page by page suggested edits.
Page 1
Please change the kind of submission from “Article” to “Case Study”. This information is found above the paper’s title, at the left margin.
“that may be fragmented along biomedical categories”—for this particular case study, was the role of behavioral health administration in addressing the COVID-19 response fragmented along biomedical categories or not? This question needs to be answered.
“This case study reflects on the implementation experiences in one jurisdiction by applying the Plan-Do-Study-Act (PDSA) improvement model”—pleases state what you mean by the PDSA improvement model, provide a reference, and explain why you adopted this model and framework.
It is at this point that the new Figure 1, mentioned above (before the page by page edits), should be inserted.
Page 3
Please remove the italics from “The DOH BHA was given less than a one-week’s notification to plan the system. Despite the short time frame, the DOH BHA gathered a team of existing partner agencies through which it already had existing contracts that could be modified and adapted for expanded scope of services for COVID response, namely, isolation and quarantine.”
Given that Figure 1 (which, with the new Figure 1 requested, will now become Figure 2) has now been moved under the subsection 2.2. Planning Tempo (which, with the request made above will now become the subsubsection 2.1.2), two things become clear. The first is that a better title for this subsubsection is “Key Planning Tasks”, as this is mentioned in the subsubsection but planning tempo is not and, furthermore, the figure is about planning tasks, rather than a planning tempo. Second, “DOH BHA’s key planning tasks included contracting facilities, the request and intake form, triage and placement criteria, coordination and workflow system, and client tracking” is an insufficient description of what the figure describes. Please mention each of the five items displayed in the workflow diagram in this sentence.
The title of Figure 1 (now to be Figure 2) should be changed to “Workflow Diagram Summary of the Key Planning Tasks of the Hawaii Isolation and Quarantine System.
Page 4
Now that Table 1 has been moved to be part of “Roles and Responsibilities”, the authors have changed the formatting of the Responsibilities column from center-justified to left justified. The MDPI template calls for center justified. What this reviewer had asked be done with this column was to move the bullet points closer to the information, i.e., reducing the tab size. The bullets still are too far away from the information—changing the justification of the column did not correct the problem the reviewer saw with the table. Please change the column back to center-justified and move the bullets closer to the text by reducing the size of the tab.
Page 5
Thank you for removing the template information. In doing so, an additional blank line was added before “3. Do” (which will now be 2.2, in the new ordering system suggested). Please remove that additional line.
In the authors response, it is stated that the change made to the paper is “Qualtrics mobile-friendly web form, an online survey, was the available survey tool”. The change that appears is instead, “Qualtrics was the available survey tool”. The change that is indicated in the authors’ cover letter is a more descriptive change. It should be the change that appears in the paper. Please change it back to what was originally intended by the authors in their cover letter.
The authors’ response to the reviewer’s question, “How many times was the form revised?” was, “Thank you for this question. The form was revised multiple times throughout the emergency period in order to best suit the needs of the operations for delivering isolation and quarantine services. The exact number of edits to individual questions, question order, adding new questions, deleting old questions was not tracked due to the continuous nature of the operations process requiring real-time feedback.” This information should be included in the text of the paper, not only answered in the cover letter. It should appear before the new information that has been inserted about Qualtrics. The information about Qualtrics should thus be in a separate paragraph.
The “Do” section (which will now be the “Do” subsection, as part of the suggested edit) is not structured to correspond with the work flow diagram of Figure 1 (which will now be Figure 2, as suggested). Each of the, now, subsubheadings should be related to the elements of Figure 1 (Figure 2). Currently, the subheadings are “Request and Intake Form”, “Client Tracking System”, and “Overall Workforce”. What the subsubheadings should be, given Figure 1 (Figure 2) are as follows: “Intake Form”, “Triage”, “Client Connection”, “Screening”, and “Discharge”. If the authors don’t consider that these new subsubheadings reflect what they want to say and the three subheadings they currently have are a better indication of what was done, then the work flow diagram should be modified to relate to the, now, subsubheadings the authors think best represent “Do”.
Page 6
As mentioned above, “Study and Act” need to be separated, as each is just as important in the PDSA model. “4. Study and Act” should be re-headed as “2.3 Study” and “2.4 Act”. Once this is done, it become clear that the majority of the information currently in “4. Study and Act” is actually only about “Study”. Only the final paragraph is really about acting and the acting that it mentions are problems related to using the technology. Once PDSA is defined, as suggested it should be above—on page 1, then it will become clear what needs to be mentioned in this section regarding “Act”. In other words, “Act” will need to be expanded in this subsection based on the definition of PDSA that is now to be provided on page 1.
Page 7
In moving Table 1, the authors added a blank line in between paragraphs, please remove the blank line.
Page 8
The “Discussions” section includes not only a discussion but also what should be the conclusions. The current “Conclusions” section is not a proper conclusion to the work that was reported in this paper. It is suggested that the authors end the discussion with the first paragraph on this page and make the “Conclusions” as follows (notice it is suggested that all the information currently in the “Conclusions” should be eliminated). The renumbering of “Conclusions” corresponds to the suggestion made above concerning the new section, “2. The Plan-Do-Study-Act Improvement Model and Framework”.
“4. Conclusion
This case study reviews the PDSA model that was utilized in Honolulu, Hawaii to implement a COVID-19 isolation and quarantine facility system with a behavioral health system, as well as lessons learned and implementation plan of this one city in the United States that were not exclusively clinical or laboratory focused.
Like many cities across the United States, Honolulu has been facing serious social challenges of homelessness and housing, mental health, and substance use, further exacerbated by the challenges of a COVID-19 pandemic and other infectious diseases, which have been a long-standing concern of Hawaii as a tropical location. There is thus a need for timely and accurate communication. This paper contributes to literature about the role of programs that address the social, economic, and psychological underpinnings that contribute to the risk of infectious disease spread.
Future research would benefit from a comprehensive review of studies and experiences about isolation and quarantine hotels around the world. In particular, future research is needed to understand the ways in which behavioral health was integrated and coordinated as part of the COVID-19 response and the ways in which organizational structures and siloes hindered or helped to integrate these two challenges. This narrative historical case study is one innovative though small contribution.”
Page 9
Although the authors claim that the references have been changed to MDPI style, they have been changed, but not to that style. Please check the template for the reference style and redo the references.
Reviewer 3 Report
All comments were addressed to provide reviewer suggestions clear in terms of scientific soundness.
Author Response
"All comments were addressed to provide reviewer suggestions clear in terms of scientific soundness"
Authors' response: Thank you so much for all of the reviewer’s detailed and very helpful comments, which we believe have greatly improved the paper!